# A Glutathione Peroxidase Gene from *Litopenaeus vannamei* Is Involved in Oxidative Stress Responses and Pathogen Infection Resistance

**DOI:** 10.3390/ijms23010567

**Published:** 2022-01-05

**Authors:** Jinquan Fan, Binbin Li, Qianming Hong, Zeyu Yan, Xinjun Yang, Kecheng Lu, Guoliang Chen, Lei Wang, Yihong Chen

**Affiliations:** 1Key Laboratory for Healthy and Safe Aquaculture/Institute of Modern Aquaculture Science and Engineering (IMASE), College of Life Science, South China Normal University, Guangzhou 510631, China; 20163602059@m.scnu.edu.cn (J.F.); liwenwub@163.com (B.L.); hongqianming@126.com (Q.H.); yanzero98@163.com (Z.Y.); yang_995@163.com (X.Y.); lcc961114@163.com (K.L.); Chenguoliang0523@163.com (G.C.); 2Southern Marine Science and Engineering Guangdong Laboratory (Zhuhai), Zhuhai 519000, China

**Keywords:** *Litopenaeus vannamei*, glutathione peroxidase, oxidative stress, *Antimicrobial peptide* genes

## Abstract

In shrimp, several glutathione peroxidase (GPX) genes have been cloned and functionally studied. Increasing evidence suggests the genes’ involvement in white spot syndrome virus (WSSV)- or *Vibrio alginolyticus*-infection resistance. In the present study, a novel GXP gene (*LvGPX3*) was cloned in *Litopenaeus vannamei*. Promoter of *LvGPX3* was activated by NF-E2-related factor 2. Further study showed that *LvGPX3* expression was evidently accelerated by oxidative stress or WSSV or *V. alginolyticus* infection. Consistently, downregulated expression of *LvGPX3* increased the cumulative mortality of WSSV- or *V. alginolyticus*-infected shrimp. Similar results occurred in shrimp suffering from oxidative stress. Moreover, LvGPX3 was important for enhancing *Antimicrobial peptide* (*AMP*) gene expression in S2 cells with lipopolysaccharide treatment. Further, knockdown of *LvGPX3* expression significantly suppressed expression of AMPs, such as *Penaeidins 2a*, *Penaeidins 3a* and *anti-lipopolysaccharide factor 1* in shrimp. AMPs have been proven to be engaged in shrimp WSSV- or *V. alginolyticus*-infection resistance; it was inferred that LvGPX3 might enhance shrimp immune response under immune challenges, such as increasing expression of AMPs. The regulation mechanism remains to be further studied.

## 1. Introduction

Glutathione peroxidases (GPXs) belong to a family of phylogenetically related oxidoreductases distributed in all living domains. They are selenium-dependent hydroperoxidase-reducing enzymes that contribute to removing lipid hydroperoxide, eliminating H_2_O_2_, reducing damage of organic hydroperoxides and so on [1]. Thus, GPXs play key roles in the oxidative defense reaction. GPXs are usually in the form of a homotetramer, with molecular weight of 76–95 kDa [2]. In general, the activity center of GPX is selenocysteine, and its activity is an index of the body’s selenium level [3].

GPXs are mainly divided into four subfamilies: cytosol GPXs, plasma GPXs, phospholipid hydroperoxides GPXs and gastrointestinal-specific GPXs [4]. Cytosolic GPXs mainly catalyze GSH, which participates in peroxidation and scavenges peroxides and hydroxyl radicals produced in cell respiration and metabolism, thus reducing the peroxidation of polyunsaturated fatty acids in cell membrane, which is very important [2]. Plasma GPXs have been shown to be involved in the removal of extracellular hydrogen peroxide and participation in the transport of GSH [5]. It has been reported that phospholipid hydrogen peroxide GPXs inhibit membrane phospholipid peroxidation [6]. Gastrointestinal-specific GPXs are only present in the gastrointestinal tract of rodents, and function to protect animals from ingestion of lipid peroxides [1].

Published research focuses on the relationship between GPXs’ antioxidant function and diseases. For example, GPXs in the brain significantly decreased during hypoxia, while the representation of lipid peroxide malondialdehyde (MDA) obviously increased, indicating that the antioxidant capacity in the brain was weakened during hypoxia [7]. The activity of GPXs decreased in patients with severe atherosclerosis, which may be an independent risk factor for atherosclerosis, suggesting an important link between the enzyme and such diseases. GPXs expression is also closely related to cardiovascular diseases, atherosclerosis, essential hypertension, myocarditis and so on [8]. There were reports that GPXs were involved in immune responses. The level of total antioxidant status and activities of antioxidant enzymes (GPXs and superoxide dismutases (SODs)) were remarkably decreased in tissue with oropharyngeal cancer, particularly in Epstein–Barr virus-positive cases [9]. Another study in mice found that GPXs protected mice from coxsackievirus-induced myocarditis [10]. In chicken, *Eimeria tenella* infection significantly activated SODs and GPXs [11].

White spot syndrome virus (WSSV) is the causative agent of white spot syndrome in shrimp, causing mass mortalities in aquaculture [12]. WSSV infection results in oxidative stress and the release of reactive oxygen species (ROS) that are harmful to cells. In *Fenneropenaeus indicus*, activities of superoxide dismutase, catalase, glutathione-S-transferase, reduced glutathione, GPXs and glutathione reductase were significantly reduced in WSSV-infected shrimp [12]. Similar results were also observed in *L. vannamei*. A significant reduction in antioxidant enzymes, such as GPXs, was found at 48 h post-infection in all tissues analyzed, which suggested that oxidative stress and tissue damage via inactivation of antioxidant enzymes in infecting shrimp caused systematic injuries [13]. Further, shrimp responded to Cd and pH stress as well as *Vibrio alginolyticus* infection [10,14]. It remains unclear whether there are distinct mechanisms for shrimp GPXs participating in immune response apart from its antioxidant activity.

In the present study, we cloned a novel *GPX* gene in *L. vannamei* (*LvGPX3*). The gene not only took part in oxidative stress response, but also was engaged in WSSV or *V. alginolyticus* infection resistance. Moreover, we found that *LvGPX3* could enhance expression of *AMPs* with immunostimulation, which revealed a possible mechanism by which *LvGPX3* was engaged in shrimp innate immunity.

## 2. Results and Discussion

### 2.1. LvGPX3 Cloning and Sequence Analysis

The cDNA sequence of *LvGPX3* was obtained from our transcriptome analysis data (accession No. SRP056233) and was confirmed by PCR and sequencing. The open reading frame (ORF) of *LvGPX3* was 651 bp, encoding a putative protein of 217 aa with a calculated molecular weight of 24.73 kDa (Appendix A). Conserved domain analysis using SMART (http://smart.embl-heidelberg.de, accessed on 10 October 2020) indicated that LvGPX3 carried a putative signal peptide at the amino end (Figure 1A). Predicted with PSORT Ⅱ program (https://psort.hgc.jp/cgi-bin/runpsort.pl, accessed on 10 November 2020), LvGPX3 was a secretory protein. Subcellular localization assay showed that eGFP-LvGPX3 was widely distributed in cytoplasm, which could be the newly synthesized LvGPX3 that had not yet been transported out of the cells (Figure 2B). The results of Western blot assay revealed that a substantial part LvGPX3 was secreted to the extracellular (Figure 2C). Moreover, LvGPX3 possessed a conserved GSHPx domain (Figure 1A), which is commonly found in GPXs and is the feature of enzymes catalyzing the reduction of hydroxyperoxides by glutathione. The catalytic site of GSHPx contains a conserved residue, which is either a cysteine, or, as in many eukaryotic GPXs, a selenocysteine. Interestingly, there is also a cysteine in the LvGPX3 GSHPx domain—whether or not it is selenocysteine still remains unknown.

### 2.2. Phylogenetic Analysis

To investigate the relationships among LvGPX3 and its homologs, a multiple sequence alignment was constructed (Appendix A). LvGPX3 was highly similar to mollusk GPXs, comparing with other arthropods’ GPXs. For example, LvGPX3 was 95.37% and 50.69% identical to *Penaeus monodon* GPX and *Pomacea canaliculata* GPX, respectively (Appendix A). A phylogenetic tree was generated using the neighbor-joining (NJ) method. In this phylogeny tree, GPXs and associated proteins fell into three subgroups (Figure 1B): Group 1 contained eight invertebrate GPXs (HsGPX, PtGPX, OoGPX, OaGPX, CmGPX, GgGXP, XlGPX and PmGPX); Group 2 included two GPXs (TcGPX and ApGPX); and Group 3 also contained eight GPXs (NlGPX, ZnGPX, LpGPX, MeGXP, PcGPX, PtrGPX, PmGPX and PmGPX). GPXs are relatively conservative; it is difficult to reveal their specific functions from their domain distribution or phylogenetic analysis. More experimental evidence is needed to discover LvGPX3’s specific roles in *L. vannamei*.

### 2.3. Constitutive Transcription and Protein Expression of LvGPX3

Real-time RT-PCR analysis indicated that *LvGPX3* was expressed in all examined tissues. It was extremely highly expressed in the shrimp intestine: *LvGPX3* expression in intestine was about 21.6-fold greater than in muscle. *LvGPX3* was also enhanced in hemocytes, which was 14.3-fold greater than in muscle (Figure 2A). As both hemocytes and intestine are important immune tissues for shrimp [15,16], this result suggested that LvGPX3 was potentially involved in *L. vannamei* immune response.

### 2.4. LvGPX3 was Induced with Oxidative Stress

To investigate the regulation mechanism of *LvGPX3*, four vectors carrying the 1500-bp *LvGPX3* promoter region or its mutants were constructed: pGL4-LvGPX3 contained the wild-type AREs; pGL4-LvGPX3mARE1, pGL4-LvGPX3mARE2 and pGL4-LvGPX3mARE contained one, one and two mutant AREs, respectively (Figure 3A). Dual-luciferase reporter gene assay indicated that over-expression of LvNrf2, which was the key transcription factor in the cells’ redox system, significantly increased pGL4-LvGPX3 activity; mutation of the *LvGPX3* promoter at each ARE sharply impaired the LvNrf2 activation effect on *LvGPX3*; finally, the activity of the double-ARE mutation *LvGPX3* promoter decreased more than that of the single-ARE mutation *LvGPX3* promoters under LvNrf2 stimulation (Figure 3B). Thus both AREs in the *LvGPX3* promoters contributed to its oxidative-stress response activity. *LvGPX3* expression in the shrimp hemocytes was detected by real-time RT-PCR assays. When shrimps were treated with *β*-glucan, *LvGPX3* expression increased from 3 h post injection, peaking at 48 h post injection (6.8-fold greater than the expression level of the control; Figure 3C). These results strongly suggested that LvGPX3 played an important role in shrimp antioxidant system. In fact, it is the basic function of the GPX-family proteins [17].

### 2.5. Knockdown Expression of LvGPX3 Depressed the Antioxidative Stress Response in Shrimp

We further investigated the antioxidant stress role of LvGPX3 in vivo by RNAi plus *β*-glucan injection, which has been proven to cause oxidative stress in shrimp [18]. At 132 h post *β*-glucan injection, cumulative mortality was 2% for the PBS plus dseGFP group, 10% for the dseGFP plus *β*-glucan group, 4% for the dsLvGPX3 plus PBS group and 20% for the dsLvGPX plus *β*-glucan group (Figure 4). Thus the downregulation of *LvGPX3* increased shrimp cumulative mortality due to oxidative stress. Considering the results of the aforementioned experiments, LvGPX3 likely has a role in maintaining the balance of redox systems in *L. vannamei* [17].

### 2.6. LvGPX3 Expression was Induced after Infection with WSSV or V. alginolyticus

*LvGPX3* expression was measured using real-time RT-PCR assays. *LvGPX3* expression in shrimp hemocytes increased from 9 h to 96 h post WSSV infection, and reached its peak value, which was about 4.8-fold higher than the control, at 36 hpi (Figure 5A). When shrimps were infected with *V. alginolyticus*, *LvGPX3* expression accelerated from 8 hpi, peaking at 48 hpi (4.7-fold greater than the expression level of the control; Figure 5B). It seems that LvGPX3 could respond to infection by DNA viruses as well as pathogenic bacteria in *L. vannamei*.

### 2.7. Knockdown of LvGPX3 Increased the Cumulative Mortality of WSSV- or V. alginolyticus-Infected Shrimps

The real-time PCR assay showed that dsLvGPX3 effectively knocked down *LvGPX3* (Figure 6A). To better understand the anti-pathogenic properties of *LvGPX3*, we measured the cumulative mortality of *LvGPX3*-knockdown *L. vannamei* with WSSV or *V. alginolyticus* infection. In the WSSV-infected plus dsLvGPX3 injection shrimp group, cumulative mortality was 24%, 50%, 66%, 76% and 84% at 36, 48, 60, 72 and 84 hpi, respectively (Figure 6B); in the WSSV-infected plus dseGFP injection shrimp group, cumulative mortality was 8%, 16%, 34%, 42% and 54% at 36, 48, 60, 72 and 84 hpi, respectively (Figure 6B). In the *V. alginolyticus*-infected plus dsLvGPX3 injection shrimp group, cumulative mortality was 54%, 73%, 78% and 78% at 8, 12, 24 and 48 hpi, respectively (Figure 6C); finally, in the *V. alginolyticus*-infected plus dseGFP injection shrimp group, cumulative mortality was 16%, 40%, 51% and 53% at 8, 12, 24 and 48 hpi, respectively (Figure 6C). These results suggested that the response of *LvGXP3* to pathogen infection was not a certain passive response, but was a positive immune response, which is beneficial for shrimp to resist diseases.

### 2.8. LvGPX3 Enhanced the Immune Response to Pathogens in Shrimp

To disclose the mechanism by which *LvGPX3* modulated shrimp innate immunity, dual-luciferase reporter assays were carried out. The results indicated that LPS could significantly enhance the promoter activity of *Pen4*, *Mtk* and *Drs* (Figure 7A). LvGPX3 plus LPS, but not eGFP plus LPS or LvGPX3 by itself, could further increase AMP activity (Figure 7A). To test this conjecture, we investigated the function of LvGPX3 in immune-challenging response in shrimp by dsRNA plus LPS injection. The results of real-time RT-PCR showed that, except *Crustin* (GenBank accession no. MG883729; Appendix A), the expression of *Penaeidin 2a* (*Pen2a*, GenBank accession no. XM_027360478.1; Figure 7B(a)), *Pen 3a* (GenBank accession no. XM_027360479.1; (Figure 7B(b)) and anti-lipopolysaccharide factor 1 (ALF1, GenBank accession no. XM_027372864.1; Figure 7B(c)) were significantly depressed compared to the control (Figure 7B). These results suggested that LvGPX3 did not activate the immune-signal pathway directly; instead, it is more likely to play an important role in maintaining or enhancing the activity of immune response in shrimp. These results reminded us of a recently published study: GPX4 facilitates stimulator-of-interferon genes (STING) activation by maintaining redox homeostasis of lipids in mammals [19]. Although there was controversy about whether interferon factors exist in shrimp, it has been reported that shrimp have a STING protein [19,20]. Whether LvGPX3 affects STING-mediated immune function in *L. vannamei* remains unknown.

## 3. Materials and Methods

### 3.1. Plasmid Vectors Construction

The *LvGPX3* promoter, which was obtained from the *L. vannamei* Genome Sequencing Project (GenBank assembly accession: GCA_002993835.1), was 1,500 bp in length. The reporter gene vector pGL4-LvGPX3 was constructed based on pGL4-Basic (Promega, Madison, WI, USA, 53701), and PCR products were amplified using the primers listed in Table 1. The antioxidant response element (ARE) mutants (−1, 065 bp~−1, 052 bp, pGL4-LvGPX3mARE1; −446 bp~−434 bp, pGL4-LvGPX3mARE2; both AREs are mutated, pGL4-LvGPX3mARE) were constructed based on pGL4-LvGPX3 using the TaKaRa MutanBEST Kit (TaKaRa, DaLian, China, 116699). The expression vector encoding *L. vannamei* NF-E2-related factor 2 (LvNrf2; 1~788 aa) or LvGXP3 was constructed using pAC5.1-Basic, and the primers were listed in Table 1. The *Mtk*, *Pen4* and *Drs* report-gene vectors were previously constructed by He’s lab [22].

### 3.2. Cell Culture and Dual-Luciferase Reporter Gene Assay

The S2 cells were cultured at 28 °C with Schneider Insect Medium (Sigma, Shanghai, China, 200031) supplemented with 10% FBS in 96-well plates for 24 h. We then transfected 50 ng of firefly luciferase reporter gene plasmid, 15 ng of pRL-TK renilla luciferase plasmid, and 50 ng of expression plasmid (the pAC5.1-Basic plasmid was used as a negative control) into each well of S2 cells using Lipofectamine 3000 Transfection Reagent (ThermoFisher, Waltham, MA, USA) following the manufacturer’s instructions. Fluorescence intensity was measured at 48 h post-transfection. LPS was added to the cells at a concentration of 5 μg/mL for a period of 6 h [22,23]. All experiments were repeated three times.

### 3.3. Synthesis of Double-Stranded RNA

The DNA templates of LvGPX3 double-strand RNA (designated dsLvGPX3) were amplified using the primer pairs DsRNA-dsLvGPX3-T7-F/DsRNA-dsLvGPX3-R and DsRNA-dsLvGPX3-F/DsRNA-dsLvGPX3-T7-R (Table 1). The PCR products were used as RNA templates, and then transcribed and purified in vitro using the RiboMAXTM Large-Scale RNA Production System-T7 (Promega, Madison, WI, USA, 53701) following the manufacturer’s protocols. DsLvGPX3 fragments were 499 bp in length. DNA templates for eGFP dsRNA (dseGFP) were prepared as previously described [23,24]. Once the concentration of the dsRNA was discovered, it was properly diluted then stored in a −80 °C refrigerator.

### 3.4. Oxidative Stress Induction and Preparation of Templates for Real-Time RT-PCR Assays

To induce oxidative stress, shrimps were injected with either 5 μg/50 μL *β*-glucan or PBS (control; *n* = 50 per group) [25]. Total RNA was quickly isolated from hemocytes at 0,3, 6, 12, 18, 24, 36, 48 and 72 h after *β*-glucan injection.

Total RNA was extracted using RNeasy Mini Kits (Qiagen, Hilden, Germany, 40721), and was reversely transcribed into cDNA using PrimeScript RT Reagent Kits (TaKaRa, DaLian, China, 116699). Real-time RT-PCR assays were performed with a LightCycler 480 System (Roche, Germany). The results were calculated using the 2*^−ΔΔC^*^t^ method, after normalization to *LvEF1a* (GenBank Accession No. GU136229).

### 3.5. Pathogenic Challenge

Healthy *L. vannamei* (~8 g) were obtained from a shrimp farm on Haiou Island, Guangzhou City, Guangdong Province, China. The shrimps were acclimated in a recirculating water tank system filled with air-pumped seawater (~3% salinity) at ~28 °C. Shrimps were allowed to acclimatize for one week before experimentation. To determine the gene expression profiles of *LvGPX3* in shrimp infected with WSSV or *V. alginolyticus*, healthy *L. vannamei* were injected intramuscularly at the second abdominal segment with 50 μL WSSV inoculum (~10^6^ virions) or with 50 μL of *V. alginolyticus* (7.0 × 10^5^ CFU/g). We collected the hemocytes of five WSSV-infected shrimps and five *V. alginolyticus*-infected shrimps at 0, 3, 6, 9, 12, 24, 30, 36, 48, 72 and 96 h post-injection (hpi), respectively; the hemocytes of five *V. alginolyticus*-infected shrimps at 0, 4, 8, 12, 24, 36, 48, 72 and 96 hpi. Five shrimps injected with PBS were used as the control group. Methods of total RNA extraction and gene detection were as described above.

### 3.6. Cumulative Mortality of LvGPX3-Knockdown Shrimp following Injection with WSSV, V. alginolyticus or β-Glucan

*LvGPX3* was downregulated by RNA interference (RNAi) mediated by sequence-specific dsLvGPX3. A real-time RT-PCR assay was performed 48 h after dsRNA injection to measure RNAi efficiency. LvEF1a was used as an internal control. To determine the cumulative mortality of *LvGPX3*-knockdown shrimps, healthy shrimps (*n* = 50 per group) were injected at the second abdominal segment with 8 μg of dsLvGPX3, dseGFP or PBS (shrimp were injected with 1 μg dsRNA per gram of body weight, and the volume of the injection was 50 μL). Approximately 48 hpi, shrimps were injected with 50 μL of WSSV inoculum or 50 μL of *V. alginolyticus* as outlined above.

To investigate the role of LvGPX3 in oxidative stress response, shrimps (*n* = 50) were injected with dsLvGPX3 (8 μg/shrimp), dseGFP or PBS. After 48 h, all shrimps were injected with 10 μg/50 μL *β*-glucan. Then the cumulative mortalities were recorded every 4 h.

### 3.7. Statistical Analysis

Numerical data were presented as mean ± standard deviation (SD). The means of two samples were compared by Student’s *t*-test. The differences were significant at *p* < 0.05 in all cases. All experiments were repeated at least thrice. In addition, the differences in mortality levels between treatments were analyzed by the *Kaplan–Meier* plot (log-rank *X^2^* test).

## 4. Conclusions

In the present study, we cloned a novel *GPX* gene (*LvGPX3*) from *L. vannamei*. *LvGPX3* responded to oxidative stress and was regulated by LvNrf2. *LvGPX3* was also induced by the shrimp pathogens WSSV or *V. alginolyticus*. Moreover, we proved that *LvGPX3* was involved in *AMP* regulation, which may contribute to its anti-pathogenic-infection activity. These results provided us with new information about GPXs in *L. vannamei*, and that the anti-pathogenic function of *LvGPX3* may depend on its *AMP* regulation ability. Further research will focus on the mechanisms by which LvGPX3 regulated *AMPs*. In addition, this research also inspired research on how to improve the antiviral function of *L**. vannamei*. The antiviral function of LvGPX3 provides a theoretical reference point for aquaculture environmental regulation, drug treatment, feed additives and other disease resistance research.

## Figures and Tables

**Figure 1 ijms-23-00567-f001:**
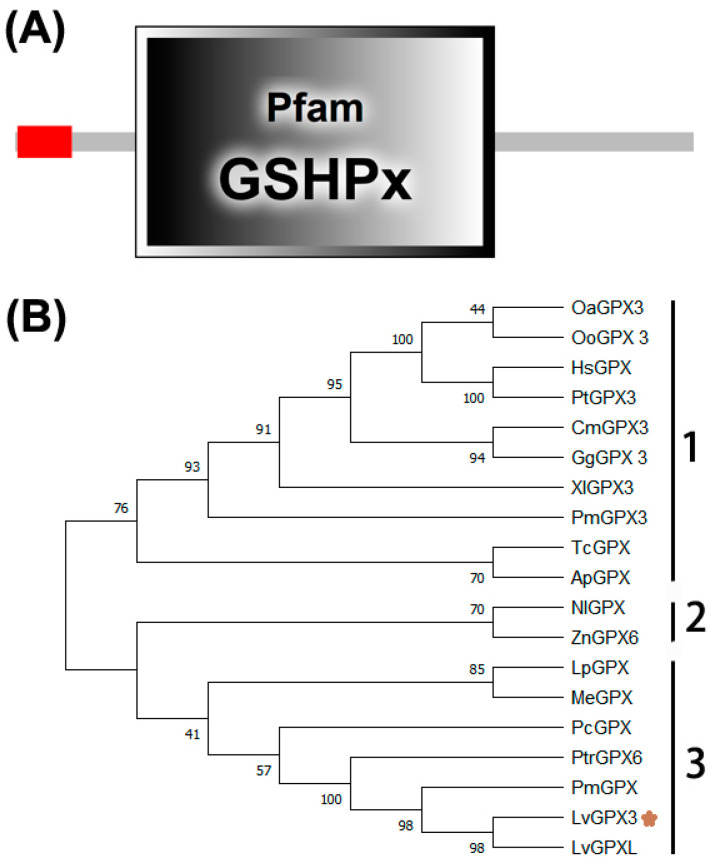
**LvGPX3 domain distribution and phylogenetic analysis of the GPXs.** (**A**) Schematic representation of the structural motifs of LvGPX3; (**B**) Phylogenetic tree of PGXs. HsGPX, Homo sapiens glutathione peroxidase (GenBank accession No. BAA00525.1); PtGPX3, *Pan troglodytes* glutathione peroxidase 3 precursor (GenBank accession No. NP_001108629.1); OoGPX3, *Orcinus orca* glutathione peroxidase 3 (GenBank accession No. XP_004280386.2); OaGPX3, *Ovis aries* glutathione peroxidase 3 (GenBank accession No. XP_014951639.1); CmGPX3, *Chelonia mydas* glutathione peroxidase 3 isoform X2 (GenBank accession No. XP_027689463.1); GgGPX3, *Gallus*
*gallus* glutathione peroxidase 3 precursor (GenBank accession No. NP_001156704.1); XlGPX3, *Xenopus laevis* glutathione peroxidase 3 homeolog precursor (GenBank accession No. NP_001085319.2); PmGPX3, *Petromyzon marinus* glutathione peroxidase 3-like (GenBank accession No. XP_032814057.1); TcGPX, *Tribolium castaneum* glutathione peroxidase (GenBank accession No. NP_001164309.1); ApGPX, *Acanthaster planci* glutathione peroxidase-like (GenBank accession No. XP_022106637.1); NlGPX3, *Nilaparvata lugens* glutathione peroxidase 3-like (GenBank accession No. XP_022184461.1); ZnGPX6, *Zootermopsis* nevadensis glutathione peroxidase 6-like isoform X2 (GenBank accession No. XP_021936526.1); LpGPX, *Limulus polyphemus* glutathione peroxidase-like isoform X1 (GenBank accession No. XP_013772703.1); MeGPX, *Metapenaeus ensis* glutathione peroxidase (GenBank accession No. ACB42236.1); PcGPX, *Pomacea canaliculata* glutathione peroxidase-like (GenBank accession No. XP_025090868.1); PtrGPX6, *Portunus trituberculatus* Glutathione peroxidase 6 (GenBank accession No. MPC16807.1); PmGPX, *Penaeus monodon* selenium-dependent glutathione peroxidase (GenBank accession No. AQW41378.1); LvGPXL, *Penaeus vannamei* glutathione peroxidase-like (GenBank accession No. XP_027224546.1).

**Figure 2 ijms-23-00567-f002:**
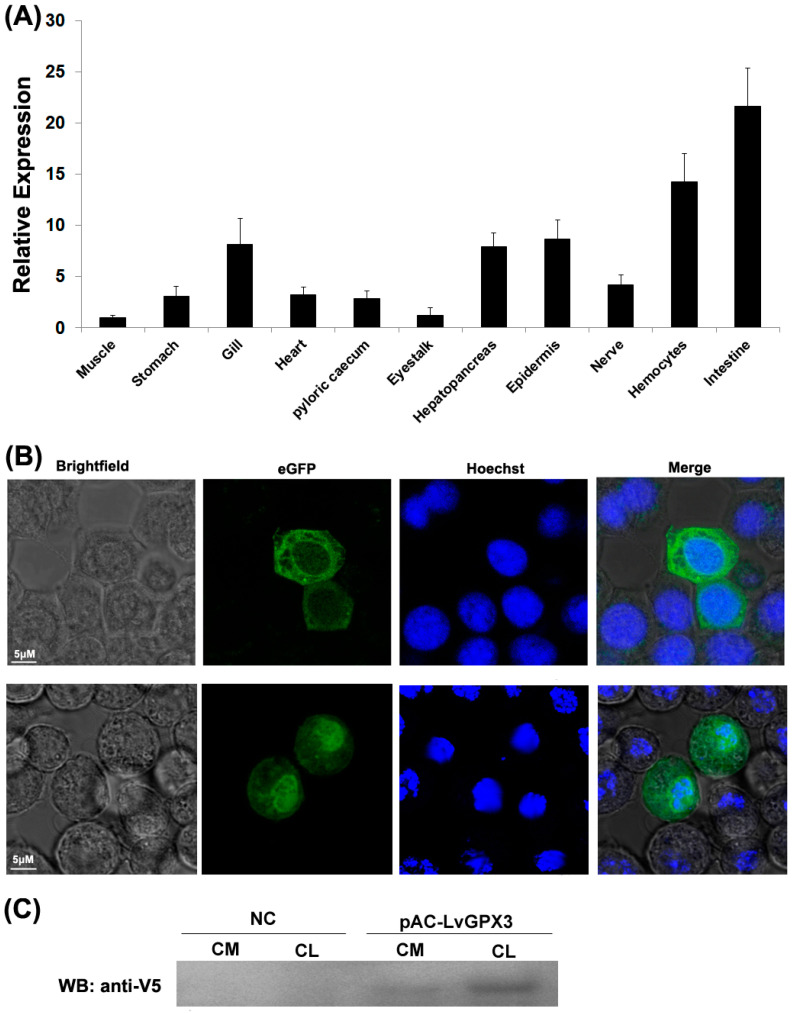
**Expression profile of *LvGPX3* and secretory expression of LvGPX3.** (**A**) Total RNA extracted from different tissues were reversely transcribed into cDNAs to serve as templates. Relative expression levels of *LvGPX3* were normalized to *LvEF1α*. The results are based on three independent experiments and expressed as mean values ±S.D. (**B**) The location of the proteins was visualized with a Leica laser scanning confocal microscope. (**C**) S2 cells were transfected with pAC-LvGPX3 expression plasmids. At 48 h post transfection, the cells were harvested and the V5-LvGPX3 was detected by Western blot assay using anti-V5 antibody. CM, (Culture media); CL, (Cell lysates).

**Figure 3 ijms-23-00567-f003:**
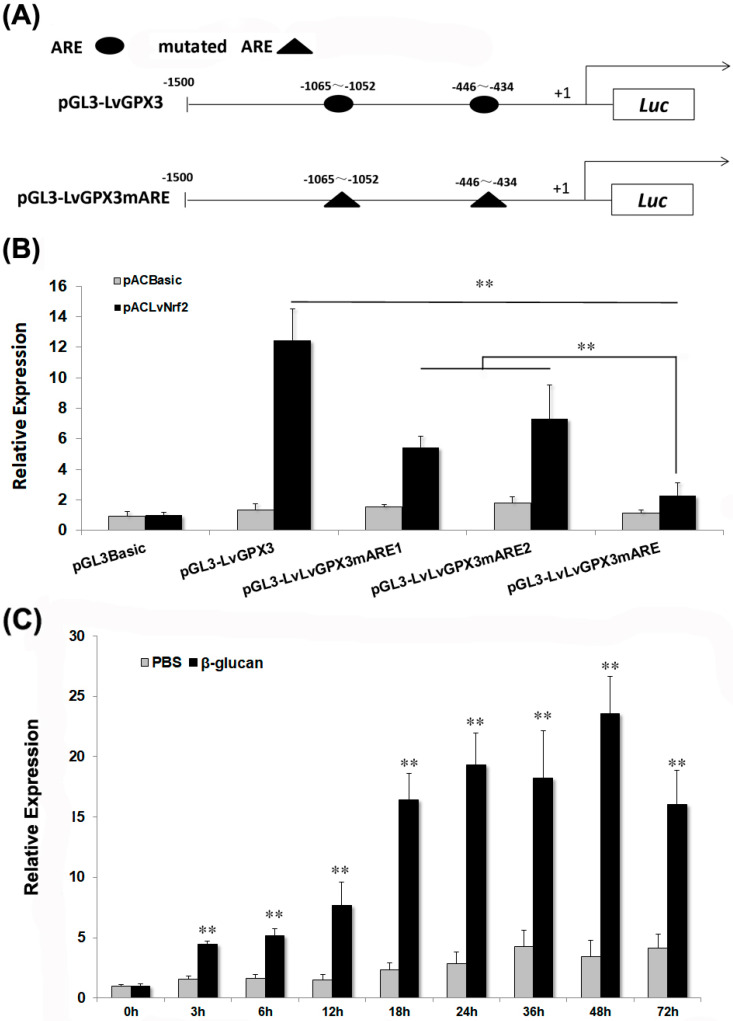
***LvGPX3* is regulated by oxidative stress response pathway.** (**A**) Schematic diagram of *LvGPX3* promoter regions in the luciferase reporter gene assay is constructed. +1 denotes the transcription initiation site for *LvGPX3* gene. Luc denotes the firefly luciferase reporter gene. The putative AREs are indicated by oval boxes. The mutated AREs are indicated with triangle boxes. (**B**) Activation of the *LvGPX3* promoter by LvNrf2. The relative luciferase activity of pGL3B-LvGPX3, pGL3B-LvGPX3mARE1, pGL3B-LvGPX3mARE2 and pGL3B-LvGPX3mARE were detected. The bars indicate mean values ±S.D. of luciferase activity (*n* = 3). (**C**) The mRNAs were collected at 0, 3, 6, 12, 18, 24, 36, 48 and 72 h after *β*-glucan injection. The relative expression of *LvGPX3* upon *β*-glucan injection was normalized with *LvEF1α* and compared to time zero. The bars represent the mean values ±S.D. of three replicates. The statistical significance was calculated using Student’s *t*-test (** indicates *p*< 0.01).

**Figure 4 ijms-23-00567-f004:**
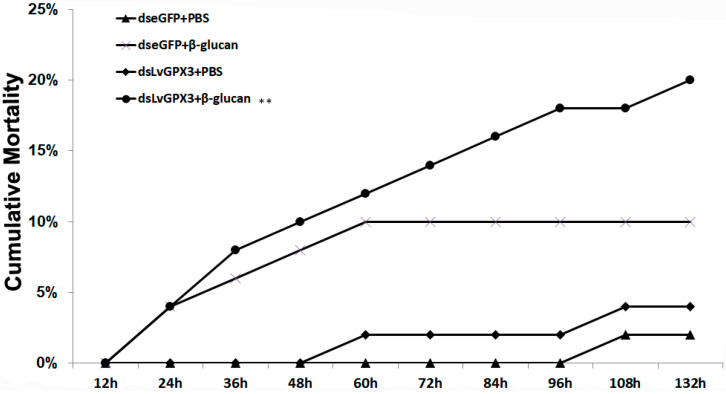
**Knockdown of expression of *LvGPX3* increased cumulative mortality of shrimps under oxidative stres****s.** Shrimp (*n* = 50) were injected intramuscularly with dseGFP (control) or dsLvGPX3. At 48 h after the initial injection, shrimps were injected with *β*-glucan or PBS. Cumulative mortality was recorded every 4 h. The mortality levels among different treatments were analyzed by *Kaplan–Meier* plot (log-rank *X^2^* test). Significant differences in *L. vannamei* mortality were marked with asterisks (** indicates *p* < 0.01).

**Figure 5 ijms-23-00567-f005:**
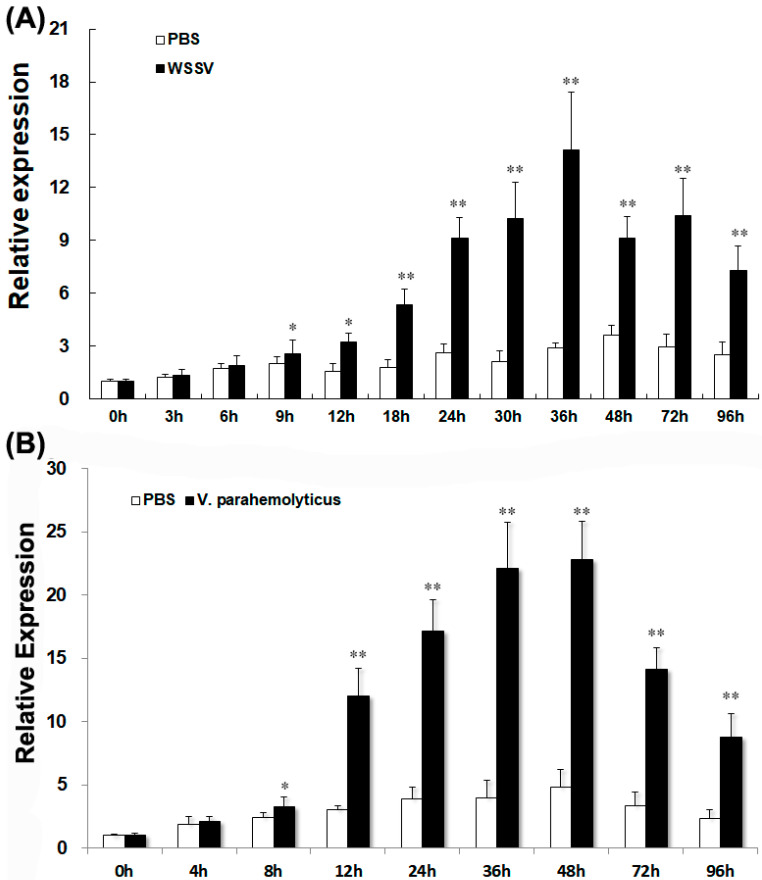
**Expression of *LvGPX3* in hemocytes of immune-challenged shrimps.** The mRNAs were collected at 0, 3, 6, 9, 12, 24, 30, 36, 48, 72 and 96 h after WSSV infection (**A**), or at 0, 4, 8, 12, 24, 36, 48, 72 and 96 h after *V. alginolyticus* infection (**B**). The relative expression of *LvGPX3* upon WSSV or *V. alginolyticus* infection was normalized with *LvEF1α* and compared to time zero. The bars represent the mean values ±S.D. of three replicates. The statistical significance was calculated using Student’s *t*-test (* indicates *p* < 0.05; ** indicates *p* < 0.01).

**Figure 6 ijms-23-00567-f006:**
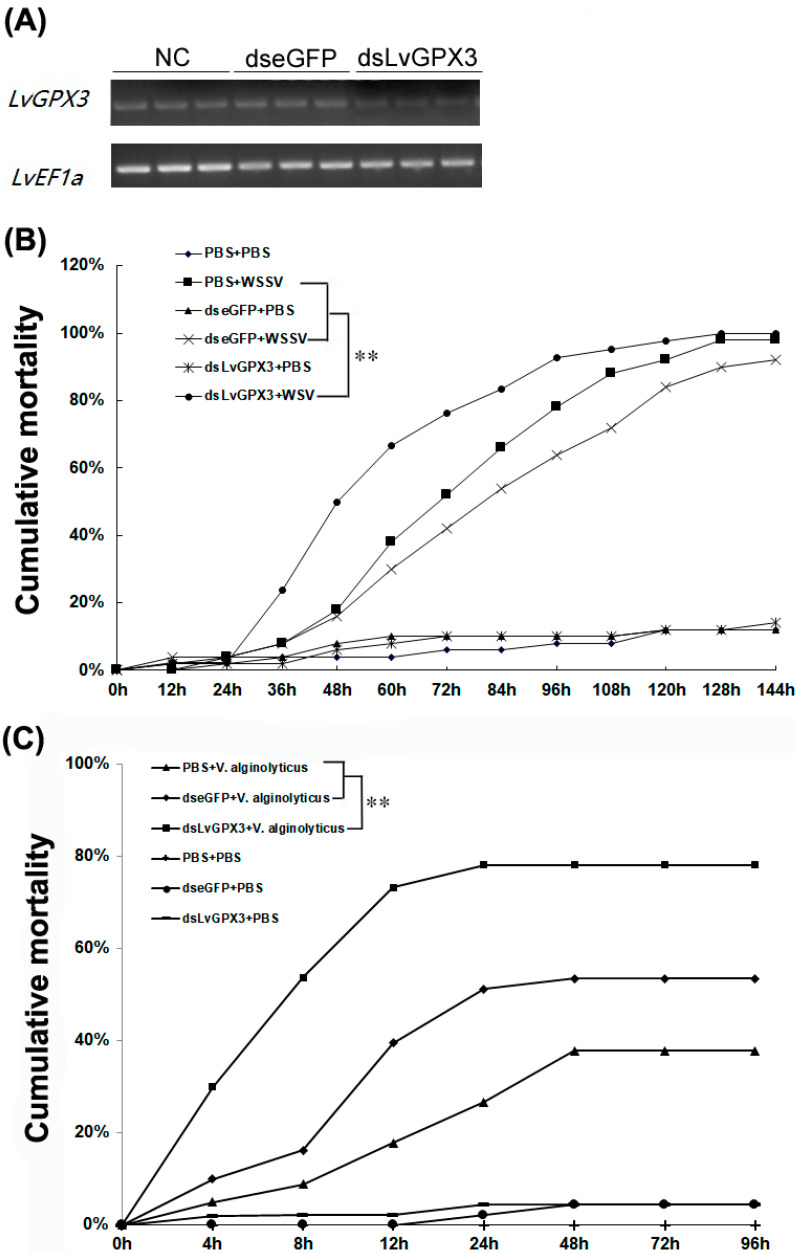
**Cumulative mortality following treatment with dsLvGPX3 plus immune challenged.** (**A**) RT-PCR analysis was carried out for determining knockdown effect. The internal control was *LvEF1α*. Samples were taken 48 h after dsRNA injection. For cumulative mortality test, shrimps (*n* = 50) were injected intramuscularly with dseGFP (control) or dsLvGPX3. At 48 h after the initial injection, shrimps were infected with (**B**) WSSV or (**C**) *V. alginolyticus*. Cumulative mortality was recorded every 4 h. Differences in mortality levels between treatments were analyzed by *Kaplan–Meier* plot (log-rank *X^2^* test). Significant differences in *L. vannamei* mortality were marked with asterisks (** indicates *p* < 0.01).

**Figure 7 ijms-23-00567-f007:**
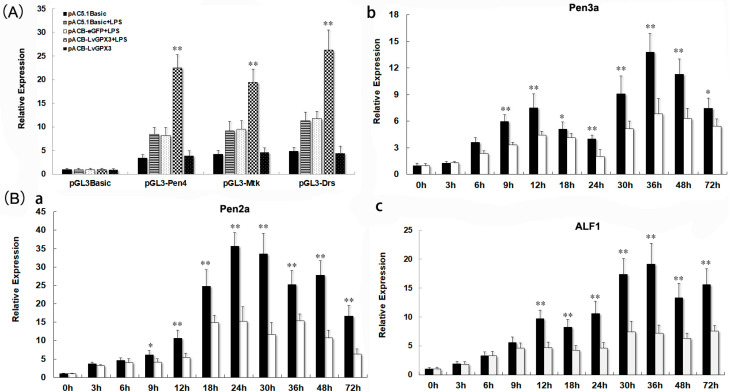
**LvGPX3 was important for enhancing *AMP* expression under immune challenge.** (**A**) The promoter activities of *Pen4*, *Mtk* and *Drs* were measured via dual luciferase reporter gene assays. LPS was added to S2 cells at a concentration of 5 μg/mL 6 h before cells were harvested. The bars indicate mean values ± S.D. of the luciferase activity (*n* = 3). (**B**) Upon LPS treatment, inhibiting LvGPX3 decreased the expression of *Pen2a* (**a**), *Pen3a* (**b**), and *ALF1* (**c**) but not *Crustin*. For LPS injection, each shrimp was injected with LPS at 36 h after dsRNA injection with a concentration of 240 μg LPS dissolved in 50 μL PBS [21]. The bars represent the mean values ± S.D. of three replicates. The statistical significance was calculated using Student’s *t*-test (* indicates *p* < 0.05, ** indicates *p* < 0.01).

**Table 1 ijms-23-00567-t001:** Summary of primers used in this study.

**Primers**	**Sequence (** **5′-3′)**
For gene expression ^a^	
pAC-LvGPX3-EcoRⅠ-F	CGG**GAATTC**TATGTTGTGGGCGGGGTTCG
pAC-LvgGPX3-XhoⅠ-R	CGG**CTCGAG**GCAAATTCCTGGCTCAGGAGGAAC
pAC-LvNrf2-EcoRⅤ-F	CGG**GATATC**ATGGAAGGCCCTGTAATTGA
pAC-LvNrf2-XbaⅠ-R	ATA**TCTAGA**CTCTGCTTGGGGTCATCCTTCCC
**For real-time RT-PCR**	
QPCR-LvGPX3-F	GATCGTCAATGTGGCGACCTA
QPCR-LvGPX3-R	CTCTTGCTTCCCGAATTGGTT
QPCR-LvPen2a-F	GGTTACAGGCCCGATACCCA
QPCR-LvPen2a-R	GTGACAACAGCTTCCGAACTTG
QPCR-LvPen3a-F	GCTTGCGTGATATGAGTGAGTG
QPCR-LvPen3a-R	AATTACAACGAAAGGCAGATGG
QPCR-LvALF1-F	GACAGGCTTCCGAGCAACAC
QPCR-LvALF1-R	GTGGCACAAGAGCAATCAGG
QPCR-LvCrustin-F	CGACGACAATGACGCAACAG
QPCR-LvCrustin-R	AAGACCTCCACCCAATCCAAA
QPCR-LvEF1a-F	GCTGATTGCGCCGTACTCAT
QPCR-LvEF1a-R	TCACGGGTCTGTCCGTTCTT
**For dsRNA template amplification**	
DsRNA-LvTRIM32-485-T7-F1	GGATCCTAATACGACTCACTATAGG TGAGCTTCAGTTTGCGTCCAG
DsRNA-LvTRIM32-485-R1	TACAGCGGGTGTTCATTCTCG
DsRNA-LvTRIM32-485-F2	TGAGCTTCAGTTTGCGTCCAG
DsRNA-LvTRIM32-485-T7-R2	GGATCCTAATACGACTCACTATAGG TACAGCGGGTGTTCATTCTCG
**For reporter gene assay ^b^**	
pGL3-LvGPX3-Kpn I-F	ATA**GGTACC**CGTACCAACAGGCAACAACA
pGL3-LvGPX3-BglⅡ-R	CCG**CTCGAG**GCTGAAAAATAAATATGAAAGGTG
pGL3-LvGPX3mARE-F1	agtcatcggatcCACGCATAAACATGCAAAAAAAC
pGL3-LvGPX3mARE-R1	TACTCCATGTGTTTACAGAAATGTAGAT
pGL3-LvGPX3mARE-F2	agacatcgcatcCACGTATAAACATAAGCATAAAAC
pGL3-LvGPX3mARE-R2	TGTTTTATGTTTATGTTTATGCGTG

^a^ Nucleotides in bold indicate restriction sites introduced for cloning; ^b^ Nucleotides in lower-case are the mutant sites.

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
