# Peer review of "A Glutathione Peroxidase Gene from Litopenaeus vannamei Is Involved in Oxidative Stress Responses and Pathogen Infection Resistance"

_ijms, 2022, doi:10.3390/ijms23010567_

Round 1
Reviewer 1 Report
In the paper entitled "A glutathione peroxidase gene from Litopenaeus vannamei is involved in oxidative stress responses and pathogen infection resistance" the authors have cloned the LvGPX3 gene from L. vannamei, involved in oxidative stress and, alco, regulated by LvNrf2. The authors proved that LvGPX3 gene was involved in AMPs regulation, which may contribute to its anti-pathogenic infection activity.
The paper is OK-ishly organized and written, therefore it could only be accepted for publication in IJMS after a minor revision:
- major English language editing (from Abstract to Conclusion);
- many words are misswritten; like on page 2 - line 74 "noly" instead of "only"; page 4 - line 171 "resluts" instead of "results", etc.;
- the references within the text have a different size compared to the text body;
- figures are too big and could be placed better within the text (eg. just after the paragraph that they are mentioned at....the paper would be easier to follow);
- figure 1 has no "A" and "B";
- "Figure legends" makes no sense on the bottom of page 4;
- figure 3 should be (A) (for (A)a), (B) (for (A)b) and (C) (for (B))....in my personal opinion, in this form is too complicated and takes too much time to understand;
- the References part should be more up to date...more than 45% of the references cited are more than 10 years old.
Author Response
Dear reviewer:
Thank you for your comments concerning our manuscript entitled “A glutathione peroxidase gene from Litopenaeus vannamei is involved in oxidative stress responses and pathogen infection resistance” (ijms-1530891). The reviewers’ suggestions and inputs were very helpful in guiding our revision of our manuscript. We have now revised the manuscript according to the reviewers’ comments. The followings are detailed responses to the reviewer’ comments.
Point 1: many words are misswritten; like on page 2 - line 74 "noly" instead of "only"; page 4 - line 171 "resluts" instead of "results", etc.;
Response 1: We are very sorry for our incorrect writing. We have carefully checked the misswritten words in the manuscript. Except for the two specific examples you mentioned("noly" instead of "only"; "resluts" instead of "results"), we have found nine misswritten words in the manuscript like on page 1- line 22 “supressed”; on page 1- line 27 “peroxidas”; on page 2- line 83 “peroxidas”, etc.And we have corrected them all. Please see line22, 27, 82, 83, 85, 104, 204, 325, 691, 729, 733 in the revised manuscript.
Point 2: the references within the text have a different size compared to the text body;
Response 2: Thank you for pointing out this problem. We have revised the size of the references within the text to be consistent with the text body.
Point 3: figures are too big and could be placed better within the text (eg. just after the paragraph that they are mentioned at....the paper would be easier to follow);
Response 3: We gratefully appreciate for your valuable suggestion. We have scaled down the figures in the text appropriately and placed them after the paragraph that they are mentioned.
Point 4: figure 1 has no "A" and "B"
Response 4: Done accordingly. Please see Fig.1 in the revised manuscript.
Point 5: "Figure legends" makes no sense on the bottom of page 4
Response 5: Thank you for pointing out this problem in manuscript. We have deleted it
Point 6: figure 3 should be (A) (for (A)a), (B) (for (A)b) and (C) (for (B))....in my personal opinion, in this form is too complicated and takes too much time to understand;
Response 6: We feel sorry for the inconvenience brought to the reviewer, and we have done accordingly. Please see Fig.3 in the revised manuscript.
Point 7: the References part should be more up to date...more than 45% of the references cited are more than 10 years old.
Response 7: Done accordingly. We have updated some references, and reduced the proportion of the references cited are more than 10 years old.

Reviewer 2 Report
The topic of the manuscript is interesting. It can be accepted after some minor amendments.
(1) Statistical protocol is missing in materials and methods section.
(2) For accumulated events such as mortality, the authors should used survival analysis.
(3) Please discuss the potential applications of your findings in agriculture.
Author Response
Dear reviewer:
Thank you for your comments concerning our manuscript entitled “A glutathione peroxidase gene from Litopenaeus vannamei is involved in oxidative stress responses and pathogen infection resistance” (ijms-1530891). The reviewers’ suggestions and inputs were very helpful in guiding our revision of our manuscript. We have now revised the manuscript according to the reviewers’ comments. The followings are detailed responses to the reviewer’ comments.
Point 1: Statistical protocol is missing in materials and methods section.
Response 1: Done accordingly. Please see please see line 711-726 in the revised manuscript.
Point 2: For accumulated events such as mortality, the authors should used survival analysis.
Response 2: We totally understand your concern. Mortality is the method we have been using before. We will seriously consider your suggestions and try to use survival analysis in future statistics.
Point 3: Please discuss the potential applications of your findings in agriculture.
Response 3: Done accordingly. Please see please see line 734-737 in the revised manuscript.
